# Perforated Hydrogels Consisting of Cholesterol-Bearing Pullulan (CHP) Nanogels: A Newly Designed Scaffold for Bone Regeneration Induced by RANKL-Binding Peptides and BMP-2

**DOI:** 10.3390/ijms23147768

**Published:** 2022-07-14

**Authors:** Cangyou Xie, Michiko Satake-Ozawa, Fatma Rashed, Masud Khan, Masaomi Ikeda, Shunya Hayashi, Shinichi Sawada, Yoshihiro Sasaki, Tohru Ikeda, Yoshiyuki Mori, Kazunari Akiyoshi, Kazuhiro Aoki

**Affiliations:** 1Department of Basic Oral Health Engineering, Graduate School of Medical and Dental Sciences, Tokyo Medical and Dental University, Tokyo 113-8549, Japan; pomelo0110@gmail.com (C.X.); ozamichy@gmail.com (M.S.-O.); fatma.rashed@hotmail.com (F.R.); mas-bhoe@tmd.ac.jp (M.K.); 2Department of Oral Pathology, Graduate School of Medical and Dental Sciences, Tokyo Medical and Dental University, Tokyo 113-8549, Japan; tohrupth.mpa@tmd.ac.jp; 3Department of Dentistry, Oral and Maxillofacial Surgery, Jichi Medical University, 3311-1 Yakushiji, Shimotsuke 329-0498, Japan; mori-ora@jichi.ac.jp; 4Department of Oral Biology, Faculty of Dentistry, Damanhour University, Damanhour 22511, Egypt; 5Department of Oral Prosthetic Engineering, Graduate School of Medical and Dental Sciences, Tokyo Medical and Dental University, Tokyo 113-8549, Japan; ikeda.csoe@tmd.ac.jp; 6Department of Oral and Maxillofacial Surgery, Graduate School of Medical and Dental Sciences, Tokyo Medical and Dental University, Tokyo 113-8549, Japan; s884osur@tmd.ac.jp; 7Department of Polymer Chemistry, Graduate School of Engineering, Kyoto University, Kyotodaigaku Katsura, Kyoto 615-8510, Japan; sawada@bio.polym.kyoto-u.ac.jp (S.S.); sasaki.yoshihiro.8s@kyoto-u.ac.jp (Y.S.); akiyoshi.kazunari.2e@kyoto-u.ac.jp (K.A.)

**Keywords:** bone regeneration, BMP-2, RANKL-binding peptide, OP3-4, cholesterol-bearing pullulan, CHPA, perforated scaffold, and nonperforated scaffold

## Abstract

The receptor activator of NF-κB ligand (RANKL)-binding peptide, OP3-4, is known to stimulate bone morphogenetic protein (BMP)-2-induced bone formation, but peptides tend to aggregate and lose their bioactivity. Cholesterol-bearing pullulan (CHP) nanogel scaffold has been shown to prevent aggregation of peptides and to allow their sustained release and activity; however, the appropriate design of CHP nanogels to conduct local bone formation needs to be developed. In the present study, we investigated the osteoconductive capacity of a newly synthesized CHP nanogel, CHPA using OP3-4 and BMP-2. We also clarified the difference between perforated and nonperforated CHPA impregnated with the two signaling molecules. Thirty-six, five-week-old male BALB/c mice were used for the calvarial defect model. The mice were euthanized at 6 weeks postoperatively. A higher cortical bone mineral content and bone formation rate were observed in the perforated scaffold in comparison to the nonperforated scaffold, especially in the OP3-4/BMP-2 combination group. The degradation rate of scaffold material in the perforated OP3-4/BMP-2 combination group was lower than that in the nonperforated group. These data suggest that perforated CHPA nanogel could lead to local bone formation induced by OP3-4 and BMP–2 and clarified the appropriate degradation rate for inducing local bone formation when CHPA nanogels are designed to be perforated.

## 1. Introduction

Bone regeneration requires three essential elements: stem cells, signaling molecules (growth and differentiation factors), and scaffold materials [1]. A receptor activator of NF-κB ligand (RANKL)-binding peptide has itself already been shown to be a novel ligand that is able to promote osteogenic signaling [2,3] and enhance bone morphogenetic protein (BMP)-2-induced bone formation both in vitro and in vivo [4,5,6,7,8,9]. A RANKL-binding peptide has been shown to be able to enhance BMP-2 signaling and the expression of osteogenic genes, leading to the promotion of osteoblast differentiation and ectopic bone formation [9], and the RANKL-binding peptide has itself been shown to stimulate bone formation in vivo [3].

In general, peptides tend to aggregate and suppress their own bioactivity [10,11,12], and the novel RANKL-binding peptide is no exception. In order to solve this problem, we previously used the cholesterol-bearing pullulan (CHP) nanogel as a scaffold of the RANKL-binding peptide [13], since CHP is known to inhibit the aggregation of the encapsulated proteins utilizing the chaperone-like function [12]. The chaperone function is an intrinsic function that can change the synthesized protein to the functionally active form by conformational modifications in the cytoplasm [14]. CHP was found to work as an artificial molecular chaperone to assist in proper folding to make active-form proteins and peptides [15]. When a RANKL-binding peptide, W9, was dissolved in neutralized buffer, it showed a large particle size, with a hydrodynamic radius of approximately 300–350 nm [13]. In contrast, when it was combined with CHP nanogel, the particle size was reduced to less than one-tenth of the original size, becoming almost the same size as the CHP nanogel, with a hydrodynamic radius of approximately 30 nm [13]. The function of the peptide, which inhibits bone resorption, also appeared when combined with CHP nanogel, while the peptide could not inhibit bone resorption without combination with the CHP nanogel [16]. Thus, CHP nanogel has been proven to prevent the aggregation of the RANKL-binding peptide and change the peptide to the active form.

CHP nanogels have already been applied in clinical trials as carriers for anticancer drugs, and their high safety has also been recognized for use in humans [17]. Recently, it was found that the acryloyl group in acrylic acid reacted with the hydroxyl groups in CHP, then self-assembled into a new functionally nanogel-crosslinking CHPA. CHPA has the property of being biodegradable by hydrolysis of the intramolecular structure, enabling complexation with proteins and sustained release [18]. A simple freeze-thaw method changed its structure to contain macrosized pores of 20–100 μm in diameter without modifying the molecular chaperone-like function of the CHPA nanogel [19]. Recently, the use of CHPA porous nanogel enabled 3D muscle regeneration, with entrapment and controlled release of cytokines, which stimulated an increase in newly-regenerated and intricately-organized myofibers [18]. However, whether or not CHPA has sufficient osteoconductive capacity remains unclear.

To use CHPA nanogels as a scaffold for localized bone regeneration in the clinical setting, it is necessary to clarify the relationship between the structure of the scaffold and the degradation rate [20], since the degradation rate of the scaffold influences the release of signaling molecules from the scaffold [21]. Several lines of evidence have shown that scaffold design and fabrication methods influence the properties of regenerated bone tissue [22,23,24]. Furthermore, the pore size and pore shape of structures were shown to influence the degradation rate and impregnated drug release [25,26]. In the same line of evidence, the addition of perforations of 300–400 μm in diameter in the scaffold material was shown to promote bone formation in comparison to other-sized perforations (106–600 μm) [27,28,29]. Recent three-dimensional (3D) printing technology has allowed precise control in designing complex scaffold structures featuring perforations of various sizes, with a high degree of porosity, and pore interconnectivity [30,31]. However, a limitation of this technology exists since there is limited compatibility with regard to fitting the scaffold material into a 3D printer and using direct injection to form the scaffold [32]. A CHPA nanogel cannot itself be directly produced using a 3D printer, as a mold is required to lead to proper polymerization of the designed scaffold. Nonetheless, the molding method for CHPA nanogel has not been established. This experiment aimed to examine whether CHPA has osteoconductive capacity when used in combination with two signaling molecules: OP3-4 and BMP-2. We also performed experiments to clarify the effects of 300–400 µm perforation in CHPA in comparison to nonperforated CHPA, using the molding method for CHPA nanogel, which has not been established.

This experiment aimed to examine whether CHPA has osteoconductive capacity when used in combination with two signaling molecules: OP3-4 and BMP-2. We also performed experiments to clarify the effects of 300–400 µm perforation in CHPA in comparison to nonperforated CHPA using the molding method.

## 2. Results

### 2.1. The Degradation Rates of CHPA Nanogels in Different pH Levels

The degradation rate of CHPA over time in buffer solutions with three different pH levels (6.0, 7.4, and 8.0) was examined. The pH level of the buffer solution influenced the rate of degradation (Figure 1). The fastest degradation rate was seen in the alkaline buffer (8.0). It started from the first day and increased rapidly, reaching 70% on day 3 then gradually slowed, reaching 80–90% on day 10. However, at neutral pH (7.4), the degradation rate started at a later point (3 days) and accelerated rapidly, reaching 60–70% on day 10. However, when the buffer solution was acidic (6.0), little degradation was observed after 10 days. We concluded that the higher the pH value, the faster the onset of degradation of CHP nanogels.

### 2.2. Perforated CHPA Had a Slower Degradation Rate than Nonperforated CHPA In Vivo

CHPA impregnated with fluorescent rhodamine was used to test the degradation rate of the two differently designed scaffolds: perforated and nonperforated CHPA. IVIS was used to detect the localization and intensity of fluorescent rhodamine for 6 weeks in vivo, as shown in Figure 2A. In the 6-week period, perforated CHPA scaffold showed a slower degradation rate than nonperforated CHPA. When compared with the same designed material group, the impregnated group with the combination of BMP-2 and OP3-4 showed a much slower degradation rate than the vehicle-impregnated group. Quantitative analyses of the fluorescent area confirmed these observations (Figure 2B). Slower degradation in the perforated scaffold also took place from three weeks after scaffold placement, and a significant difference between the perforated and nonperforated scaffolds was continuously maintained until the end of the experiment, especially in the OP3-4/BMP-2 combination group (Appendix A).

### 2.3. Changes in the Design of the Scaffold Influenced Its Osteoconductive Ability

Changes in the calcified tissue in the calvarial defect model were tracked over a 6-week period, following placement of two differently designed scaffolds: perforated and nonperforated CHPA. In vivo µCT showed that the newly calcified tissue was mostly minor, but relatively greater was the calcified tissue in the defect, observed in the OP3-4/BMP-2 subgroup at 6 weeks (Figure 3A). The newly formed calcified tissue area in the calvarial defect measured from µCT images in the perforated scaffold group was significantly higher in comparison to the nonperforated scaffold groups when the subgroups impregnated with BMP-2 and OP3-4 were compared at 3 and 6 weeks (Figure 3B). Furthermore, at 3 and 6 weeks, only the perforated scaffold group showed a higher calcified tissue area in the subgroup impregnated with BMP-2 and OP3-4 in comparison to the Vehicle CHPA alone group.

### 2.4. Perforated CHPA Scaffold Enhanced the Bone Mineral Content in Comparison to Nonperforated CHPA

To investigate the changes in the 3D bone mineral density (BMD) and bone mineral content of the newly formed bone tissue after the placement of two differently designed scaffolds, perforated and nonperforated CHPA, we performed pQCT analyses to compare the newly formed bone tissue in calvarial bone defects (Figure 4A).

In the quantitative analyses, only the perforated scaffold group showed higher total BMD in the subgroup impregnated with BMP-2 and OP3-4 in comparison to the CHPA alone or OP3-4 only subgroups. The bone mineral content in nonperforated scaffold impregnated with BMP-2 and OP3-4 was higher than that in the CHPA alone or BMP-2 only subgroups. However, in the perforated scaffold, significant differences were observed between all groups. Finally, the perforated scaffold impregnated with BMP-2 and OP3-4 showed higher bone mineral content than the nonperforated scaffold. Significant differences in cortical bone content were observed between all subgroups and the perforated scaffold impregnated with BMP-2 and OP3-4 showed higher cortical bone content than the nonperforated scaffold (Figure 4B).

### 2.5. Osteoblast Activity in the Perforated Scaffold Group Was Higher than That in the Nonperforated Scaffold Group

Histological analyses were performed to clarify the functional role of the perforated and nonperforated CHPA scaffold on osteoblast activity using undecalcified sections. Fluorescent labeling was similar at the outer periosteal site of the bone edge (Figure 5A). However, high fluorescent labeling was observed in the inner cortical bone in perforated scaffolds impregnated with both BMP-2 and OP3-4. The bone histomorphometric analyses revealed that the mineral apposition rate (MAR), which represents osteoblast activity, was significantly higher in the perforated CHPA scaffold impregnated with OP3-4/BMP-2 than in the nonperforated CHPA scaffold. Similar results were observed in the bone formation rate (BFR), which represents the amount of newly formed bone generated between day 24 to day 16 before euthanization in the combined BMP-2 and OP3-4 group (Figure 5B).

## 3. Discussion

The purposes of this study were first, to determine whether CHPA, a novel scaffold material, has local osteoconductive activity when combined with two signaling molecules, BMP-2 and RANKL-binding peptide; and second, to clarify the effects of 300–400 µm perforation in CHPA scaffold in comparison to nonperforated scaffold.

We first examined the differences in the degradation rate of CHPA nanogels in different pH buffers, pH 8.0, 7.4, and 6.0, in vitro (Figure 1). Alkaline medium showed rapid degradation from day 1, reaching 70% by day 3. While in a neutral medium, degradation began on day 3 and rapidly increased to 70% by day 8. Remarkably, in an acidic medium, little degradation occurred during the 10-day observation period. This can be beneficial because the site of injury is usually acidic due to the presence of an inflammatory microenvironment then changes to neutral with healing [33,34], so the scaffold can withstand the inflammatory phase and release the signaling molecules at the right time promoting osteoblast differentiation and the formation of regenerated bone.

We then compared the degradation rate of two differently designed scaffolds: perforated and nonperforated CHPA scaffolds. Images of the in vivo use of IVIS showed the fluorescent sites and intensities of rhodamine-bound scaffold material at 1 and 6 weeks (Figure 2A). Perforated CHPA seems to have a slower degradation rate than nonperforated CHPA scaffolds, since slower degradation was observed from three weeks after scaffold placement (Appendix A). Even though the contact surface between the scaffold and biological tissue was increased by adding perforation, the freeze-dry fabrication method increased gel density at the contact surface and made it difficult to degrade [35]. Furthermore, the addition of a fibronectin coat added at the final stage of CHPA fabrication protected the scaffold material from degradation by esterases and other enzymes [19]. This could also be the reason for the lower degradation rate in both the perforated and nonperforated OP3-4/BMP-2 subgroups in comparison to the vehicle-impregnated subgroups since two additional reagents, BMP-2 and the peptide, were coated on top of the fibronectin-coated surface. However, further investigation of the slower release behavior of signaling molecules from CHPA is needed to elucidate how the slow degradation of CHPA is linked to bone formation.

The perforated CHPA scaffold led to greater cortical bone content and a greater bone formation rate in the OP3-4/BMP-2 subgroup in comparison to nonperforated CHPA, though the degradation rate was slower in the perforated group (Figure 2, Figure 3, Figure 4 and Figure 5). On the other hand, the bone content and bone formation rate of the BMP-2 subgroups did not differ to a statistically significant extent between the perforated and nonperforated groups (Figure 3, Figure 4 and Figure 5). We assumed that slower degradation might support the sustainable release from the scaffold. In particular, the sustainable release of the OP3-4 peptide is necessary to obtain continuous stimulation of bone formation since the RANKL-binding peptide was shown to stimulate osteoblast differentiation in the later stage of differentiation as well as the early phase of differentiation in vitro [6]. On the other hand, BMP-2 is known to recruit mesenchymal stem cells in an early stage of bone regeneration. The degradation rate of both perforated and nonperforated scaffolds was similar at 2 weeks after scaffold placement (Appendix A), suggesting a similar release of BMP-2 in 2 weeks. Furthermore, the recruitment of mesenchymal stem cells induced by BMP-2 might be similar in number in both scaffolds. Since BMP-2 does not stimulate the late phase of osteoblast differentiation [36], the beneficial effects of the perforated scaffolds on bone formation seem to appear due to the sustainable slower release of OP3-4 rather than due to that of BMP-2.

The presence of perforation in the scaffolds was reported as an inducer of faster cell migration that accelerates bone growth [37]. Another line of evidence showed that a scaffold with perforations of 300–400 µm in diameter was optimal, in terms of the size of perforation, for promoting angiogenesis and increasing bone formation [38]. A similar benefit of perforation in scaffolds was also reported as a promoter of angiogenesis around the scaffold material, which is necessary for bone regeneration [39]. The perforated CHPA may have exhibited better osteoconductive capacity due to increased angiogenesis and increased mesenchymal cell population in comparison to nonperforated CHPA in terms of bone mineral density, bone mineral content, and osteogenic activity (Figure 4 and Figure 5). Regarding the difference between the single-agent subgroup and the combination of two signaling molecules, it was only evident in the perforated scaffold and not in the nonperforated scaffold. Further studies are necessary to clarify why the beneficial effects of perforation in scaffolds (e.g., increased vascularization and faster cell migration) were only exerted in the OP3-4/BMP-2 subgroup.

The enhancement of osteoblast differentiation and bone formation could be explained as follows: runt-related transcription factor 2 (Runx2) is an essential transcription factor for osteoblastogenesis [40]. Downstream signaling of BMP-2 is known to lead the enhancement of the Runx2 expression [41]. When BMP-2 binds to the type 2 receptor, the kinase activity of the type 2 receptor activates the type 1 receptor. This activation leads to the phosphorylation of Smad 1/5, then the Smad complex associated with Smad 4 is transferred to the nucleus, upregulating the expression of Runx2. On the other hand, the downstream signaling of the RANKL-binding peptide is also known to lead the enhancement of the Runx2 expression [2]. We have shown that the proline-rich domain (PRD) in the intracellular region of RANKL plays a pivotal role in the downstream signaling stimulated by RANKL-binding molecules, such as RANK-containing vesicles secreted from mature osteoclasts and the RANKL-binding peptide [2]. The PRD of the intracellular region of RANKL is associated with Src-family kinases (SFKs), then SFKs lead the signaling of the phosphatidylinositol 3-kinase (PI3K)-Akt-mammalian target of rapamycin complex 1 (mTORC1) axis, which has already been reported as one type of Runx2-stimulatory signaling through ribosomal protein S6 kinase beta-1 (S6K1) phosphorylation. Since both BMP-2 and the RANKL-binding peptide signaling activate the expression of Runx2, both ligands could be thought to accelerate osteoblast differentiation synergistically.

Taken together, our data suggest that the addition of 300–400 µm perforations in CHPA nanogels, which were used as a scaffolding material in this experiment, can enhance bone formation. Furthermore, the best osteoconductive capacity was observed when the scaffold was impregnated with the RANKL-binding peptides OP3-4 and BMP-2.

## 4. Materials and Methods

### 4.1. Scaffold Materials

#### 4.1.1. Synthesis of Acryloyl Group-Modified CHP-Rh (CHPA-Rh)

CHPA was synthesized as previously described [18]. Briefly, CHP was dissolved in super dehydrated dimethylsulfoxide (DMSO), followed by the addition of 4-(4,6-dimethoxy-1,3,5-triazin-2-yl)-4-methyl-morpholinium chloride (DMT-MM). Then, N,N-diisopropylethylamine, and acrylic acid (DIPEA) were added, and the mixture was stirred for 22 h. After stirring, the liquid was collected and dialyzed with MilliQ water, and DMSO for 4 days. The degree of substitution of the acryloyl groups in the CHPA was 9.3 acryloyl groups per 100 glucoside units (Appendix A). DMT-MM and DIPEA were purchased from Tokyo Chemical Industry Co., LTD (Tokyo, Japan). Dehydrated DMSO were purchased from Wako Pure Chemical Industries Ltd. (Osaka, Japan).

#### 4.1.2. Molds for In Vivo Scaffold Material Fabrication

Molds were designed using Geomagic Freeform^®^ (3D, systems Corporation, Burbank, CA, USA) and then printed by the photo-polymerization 3D printer using cara Print 4.0 (Kulzer GmbH, Hanau, Germany). After printing, they were cleaned with isopropanol for 10 min using cara Print Clean (Kulzer GmbH, Hanau, Germany), then photo-polymerized using HiLite Power 3D (Kulzer GmbH, Hanau, Germany). Two types of molds were prepared for scaffolds without pores and with pores of 300–400 µm in diameter and 0.5 mm in thickness. All molds were sterilized by immersion in 70% ethanol for at least 24 h. The reason for our decision regarding the perforation size was as follows. A scaffold with perforations of 300–400 µm in diameter showed higher alkaline phosphatase (ALP) activity and the increased protein expression of osteocalcin, as detected by immunohistochemistry, in comparison to the 106–212 µm-perforation group [42]. Histologically, a scaffold with perforations of 300–500 μm in diameter showed more newly formed bone along the inner surface of the perforations and enhanced angiogenesis in comparison to the 500–600 μm perforation group [28]. Furthermore, scaffold materials with 300 µm perforations had higher bone formation and the highest maximum compressive strength in comparison to the 50, 100, 500 µm-perforation groups [43].

#### 4.1.3. Preparation of the NanoCliP Gel and NanoCliP Freeze-Dry (FD) Gel

The NanoCliP gel was prepared by Michael addition of the acryloyl groups of CHPA-Rh to the thiol groups of PEGSH. Briefly, CHPA-Rh was dissolved in PBS (pH 7.4). Separately, PEGSH was dissolved in PBS and added to the CHPA-Rh solution. The mixture was poured into a previously prepared disk-shaped mold (described above) and dropped between polytetrafluoroethylene membranes and incubated at 37 °C for 1 h to obtain a disk-shaped nanogel-crosslinked hydrogel. The molar ratio of acryloyl groups to thiol groups was 2:1. The NanoClik gel was then transferred to a polystyrene dish and was frozen at −28 °C for 30 min, and thawed at 25 °C. The NanoCliP FD gel was prepared by freezing the NanoCliP gel using liquid nitrogen and then vacuum dried for 24 h. The freeze–thaw method was applied to make it perforated [19,44]. Then, the gel was impregnated with fibronectin (Human Plasma, Wako, Osaka, Japan) to give cell adhesive properties [18]. 

### 4.2. Degradation Rate analysis of CHPA in Different pH Levels In Vitro

The degradation behavior of nanogels was analyzed. Nanogels were immersed in three buffer solutions of pH 6.0, 7.4, and 8.0 at a volume ratio 100 times greater than that of the nanogels, and samples were collected over time up to day 10. The amount of degraded nanogels was determined from the supernatant after centrifugation at 3000× *g*. Quantification of degraded nanogels was performed by the phenol sulfuric acid method (the evaluation of the amount of CHPA as the amount of glucose) using a UV-visible spectrophotometer (using visible light at 490 nm; Cary 60 UV-vis, Agilent, Santa Clara, CA, USA), as described previously. The degradation rate was calculated as the percentage of the amount of nanogel released after immersion in buffer solution divided by the amount of nanogel that was initially immersed.

### 4.3. In Vivo Experimental Design

Thirty-six, five-week-old male BALB/c mice (Sankyo Lab Service, Tokyo, Japan) were used. The mice were anesthetized before surgery with subcutaneous injections of medetomidine hydrochloride (0.5 mg/kg, Domitor; Zenoaq, Fukushima, Japan) and ketamine hydrochloride (50 mg/kg, Ketalar; Sankyo, Tokyo, Japan), as previously described [6]. A calvarial bone defect was created using a biopsy punch with a diameter of 3.5 mm, the cranial bone was removed and the aforementioned CHPA scaffold with different reagent(s) described below was implanted in the bone defect. The mice were divided into two groups: a group without perforation and a group with perforation. Each group was further divided into 4 subgroups (*n* = 4) according to the supplementary reagents added to the scaffold. A blank control group (*n* = 4) with a calvarial bone defect without implantation of scaffold material was prepared. The experimental groups are summarized as follows (Figure 6):Blank control group without a scaffold material;Scaffold material + Vehicle (2 µL of LF6 buffer + 5 µL of DMSO);Scaffold material + BMP-2 (2 µg, Bioventus LLC, Durham, NC, USA);Scaffold material + OP3-4 (0.66 mg, Atlantic Peptides LLC, Concord, NH, USA);Scaffold material + BMP-2 (2 µg) + OP3-4 (0.66 mg).

The amounts of reagents used in this experiment were calculated as previously described [7]. Briefly, BMP-2 was adjusted and diluted to 1 µg/µL in LF6 buffer (5 mM monosodium glutamate, 2.5% glycine, 0.5% sucrose, 0.01% polysorbate 80, pH 4.5). OP3-4 was dissolved in DMSO to 112 µg/µL. The mice were euthanized on day 42 after surgery and scaffold placement.

### 4.4. Radiographic Assessments

A fluorescence and emission imaging system (IVIS Imaging System; Lumina XRMS Series III PerkinElmer, Waltham, MA, USA) was used to visualize the fluorescent sites emitted by the scaffold material labeled with rhodamine in vivo, and to visualize the fluorescence sites every week after scaffold placement. The fluorescence of rhodamine was excited at wavelengths: 560/620 nm; color scale: minimum 2.55 × 10^9^, maximum 5.17 × 10^9^. The fluorescent area was measured using the ImageJ software program (version 1.50 i; NIH, Bethesda, MD, USA). The percentage degradation of scaffold material was calculated as the percentage of the fluorescent area at week 1 as 100%, subtracted from the percentage of the fluorescent area at week 6 of that at week 1. The same fluorescence threshold was applied to all images as the area of scaffold material.

In vivo microfocal computed tomography (µCT: R_mCT2; Rigaku, Tokyo, Japan) was performed at 90 kV, 160 μA, and FOV 20 to obtain three-dimensional bone images of each mouse every week after scaffold placement. Calvarial bone changes in the bone defect area were measured using the ImageJ software program (version 1.50 i; NIH, Bethesda, MD, USA). The outer side of a 3.66 mm diameter circle on a two-dimensional image was set as the region of interest (ROI) and calculated.

The bone mineral content and bone mineral density, as well as the cortical bone content of the newly formed bone, were measured by peripheral quantitative computed tomography (pQCT) (XCT Research SA+; Stratec Medizintechnik GmbH, Pforzheim, Germany). The ROI was 2.8 mm × 0.7 mm, and a frontal section including the calvarial defect was set and measured. Bone density values of > 690 mg/cm^3^ were recognized as cortical bone.

### 4.5. Histological Assessments and Bone Histomorphometry

To measure osteogenic activity, calcein (Sigma-Aldrich, St. Louis, MO, USA) was injected subcutaneously twice on days 32 and 8 before euthanization, alizarin (Dojindo, Kumamoto, Japan) was injected on day 24 before euthanization, and demeclocycline (Sigma-Aldrich, St. Louis, MO, USA) was injected on day 16 before euthanization, Figure 6. All fluorescent reagents were administered at a dose of 20 mg/kg (dose solution volume; 0.1 mL/10 g body weight). After euthanization, the head was dissected and preserved in formalin buffered saline for one day then changed to PBS. The calvarial bone was cut just anterior to the defect using a saw microtome (SP1600; Leica Biosystems, Nussloch, Germany) under a constant flow of water. The bone was then embedded in SCEM compound (Section-Lab Co. Ltd., Hiroshima, Japan) and frozen into blocks at −100 °C in a freezing machine (UT2000F; Leica Microsystems, Tokyo, Japan). Undecalcified frontal sections of 5 µm in thickness were prepared using a microtome (CM3050sIV; Leica Biosystems) (as shown by the dotted line in the left lower panel of Figure 4A) and then were retrieved by adhesive Kawamoto film (Cryofilm type 2C, Section-Lab, Co., Ltd.). The fluorescence-labeled sections were observed under a fluorescence microscope and measured using the affiliated software program (FSX100; Olympus, Tokyo, Japan). The mineral apposition rate (MAR) was calculated by measuring the inter-labeled distance between an alizarin-labeled line (red color) and a demeclocycline-labeled line (yellow color) and divided by the 8-day interval. The bone formation rate (BFR) (area reference) was calculated by multiplying the mineralizing surface (MS) by the MAR. The ROI size was 100 µm square and the center of the ROI was placed at 170 µm from the periosteal side and 250 µm from the bone edge, which was close to the sagittal suture.

### 4.6. Statistical Analysis

SPSS Version 27.0 for Windows (IBM Corp, Armonk, NY, USA) was used for statistical analysis of quantitative data of the bone regeneration area, scaffold material loss rate, and osteogenic activity. Each dataset was tested with the Wilcoxon rank sum test for comparisons between two groups and with the Dunn’s test for comparisons between three or more groups. The risk rate was corrected by Bonferroni’s method. Data were expressed as the mean ± standard deviation. *p* values of < 0.05 were considered to indicate a statistically significant difference.

Sample size calculations were conducted using the formula described below:

N = 2 (Eα/2 + Eβ) ^2^ SD^2^/φ^2^: the minimum number of specimens that were expected to obtain a significant difference in osteogenic activity between the scaffold-only group and the scaffold plus OP3-4 + BMP-2 combination group; Eα/2 = 1.96: fixed number in the case of significance = 5%; Eβ = 1.282: fixed number in the case of power = 90%; SD: standard deviation; φ: difference between the mean value; Aν1 and Aν2:N = 2 × (1.96 + 1. 282)^2^ × (SD)^2^/(Aν1 − Aν2)^2^
3.74 = 2 × (1.96 + 1.282)^2^ × (0.43)^2^/(1.02)^2^

The standard deviations and mean differences (SD = 0.43, Aν1 − Aν2 = 1.02) of the mineral apposition rate (MAR) and osteogenic activity were obtained from the results of a previous study where similar comparisons were performed [6].

## 5. Conclusions

In conclusion, we herein demonstrated that CHPA nanogel is a novel scaffold that could be applied for local bone regeneration in a murine calvarial bone defect model when RANKL-binding peptides, OP3-4 and BMP-2, were used as signaling molecules to stimulate bone formation. Furthermore, the newly designed 300–400 µm-perforations of CHPA scaffolds using 3D-printed molds could be a more appropriate structure for local bone formation induced by OP3-4 and BMP-2 in comparison to a nonperforated scaffold.

## Figures and Tables

**Figure 1 ijms-23-07768-f001:**
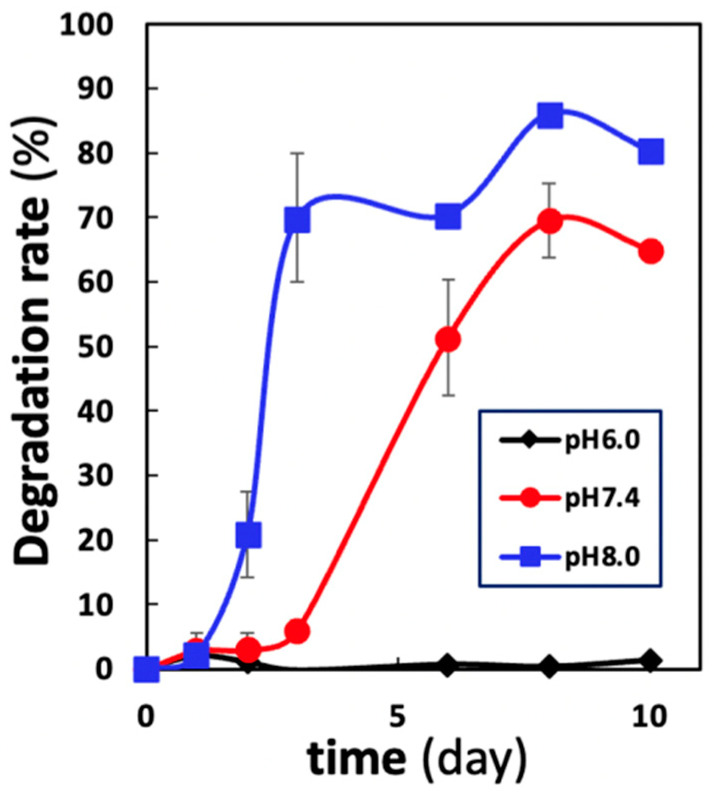
Time–course changes in the degradation rate of CHPA nanogels in different pH buffers. The degradation rate of CHPA nanogels in different pH buffers (pH 8.0, 7.4, and 6.0). Nanogels were immersed in three buffer solutions of pH 6.0, 7.4, and 8.0 at a volume ratio 100 times greater than that of the nanogel, and samples were collected over time up to day 10. The degradation rate was calculated as the percentage of the amount of nanogel released after immersion in the buffer solution divided by the amount of nanogel that was initially immersed.

**Figure 2 ijms-23-07768-f002:**
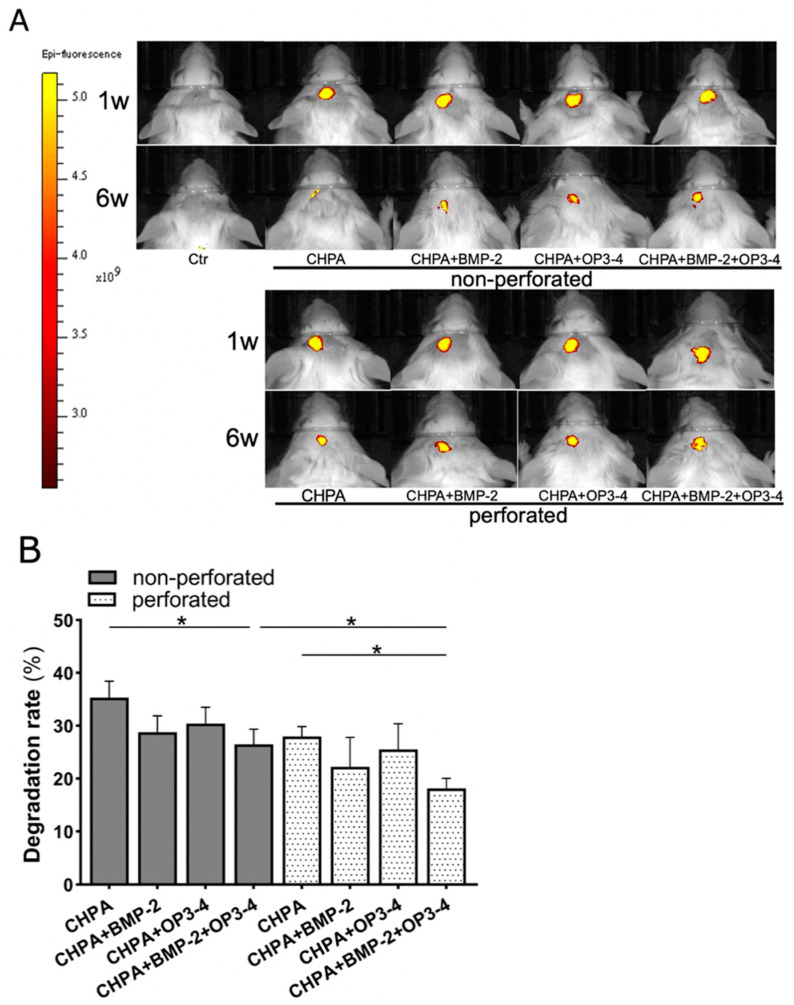
Perforated CHPA scaffold showed a slower degradation rate than nonperforated CHPA: (**A**) In vivo images of IVIS. Fluorescent sites and intensities of rhodamine-bound scaffold material at 1 and 6 weeks, after scaffold placement. Yellow areas in the figure represent the location and fluorescence intensity of rhodamine-bound CHPA. Excitation/absorption wavelengths: 560/620 nm; color scale: minimum 2.55 × 10^9^, maximum 5.17 × 10^9^. Ctr: control group; CHPA: Scaffold material + Vehicle; CHPA + BMP-2: Scaffold material + BMP-2; CHPA + OP3-4: Scaffold material + OP3-4; CHPA + BMP-2 + OP3-4: Scaffold material+BMP-2 + OP3-4. (**B**) Quantitative data of the degradation rate of the two differently designed scaffolds. The percentage reduction of scaffold material was calculated by taking the fluorescent area at week 1 as 100% and subtracting the fluorescent area percentage at week 6, as described in the Materials and Methods. Normality was analyzed by the Shapiro–Wilk test. Comparisons between groups were performed by the Wilcoxon rank sum test. Values are expressed as the mean ± SD, * *p* < 0.05.

**Figure 3 ijms-23-07768-f003:**
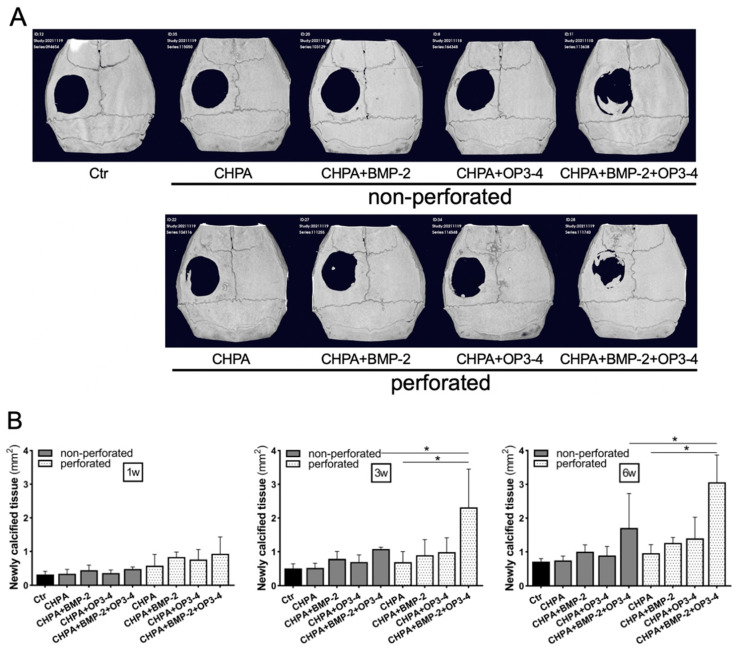
Perforated CHPA scaffold increased the area of newly calcified tissue in the calvarial defect model: (**A**) In vivo µCT images of newly calcified tissue in the calvarial defect model at 6 weeks after scaffold placement. See the Figure 2 legends for group names. (**B**) The newly calcified tissue area in the calvarial defect. The newly formed calcified tissue area was quantified at 1, 3, and 6 weeks after scaffold placement. Normality was analyzed by the Shapiro–Wilk test. Comparisons between groups were performed using the Wilcoxon rank sum test. Values are expressed as the mean ± SD, * *p* < 0.05.

**Figure 4 ijms-23-07768-f004:**
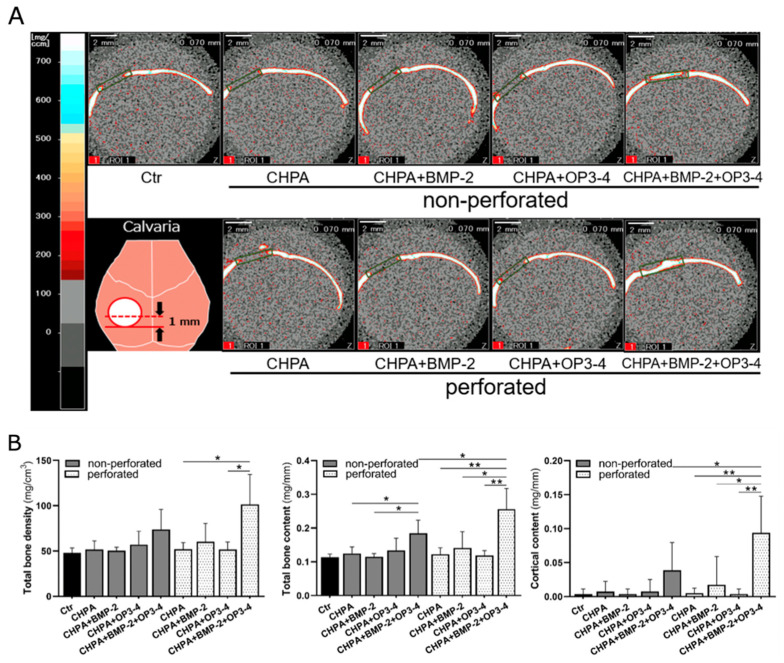
pQCT confirmed the higher bone mineral content in the perforated scaffold group: (**A**) pQCT images of newly formed bone in the calvarial defect model. The dotted line in the diagram of the lower-left panel illustrates the measurement site. Frontal section images at the site of the dotted line are shown. The green square indicates the region of interest (ROI) where bone density and other parameters were measured. Scale bar = 2 mm; See the Figure 2 legend for group names. (**B**) Quantitative data of the pQCT at the ROI of the bone defect site. Total bone density: total bone mineral density; Total bone content: total bone mineral content; Cortical area: cortical bone area. Normality was analyzed by the Shapiro–Wilk test. Comparisons between groups were performed using the Wilcoxon rank sum test. Values are expressed as the mean ± SD, * *p* < 0.05, ** *p* < 0.01.

**Figure 5 ijms-23-07768-f005:**
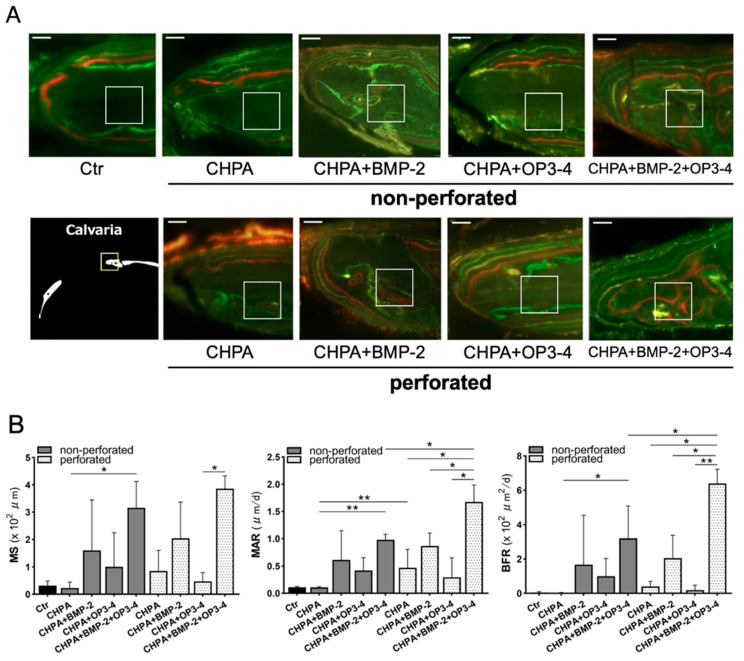
The perforated CHPA scaffold group showed higher bone formation in comparison to the nonperforated scaffold group: (**A**) Fluorescence micrographs of undecalcified sections at the bone edge of the defect site. The white 100-μm square inset represents the ROI for measurement. Green fluorescence indicates calcein administered on days 8 and 32 before euthanization; red fluorescence, alizarin administered on day 24 before euthanization; and yellow fluorescence, demeclocycline administered on day 16 before euthanization. Scale bar = 50 μm (**B**) Quantitative analyses of bone histomorphometry. Mineralizing surface (MS), mineral apposition rate (MAR), bone formation rate (BFR). Normality was analyzed by the Shapiro–Wilk test. Comparisons between groups were performed using the Wilcoxon rank sum test. Values are expressed as the mean ± SD, * *p* < 0.05, ** *p* < 0.01.

**Figure 6 ijms-23-07768-f006:**
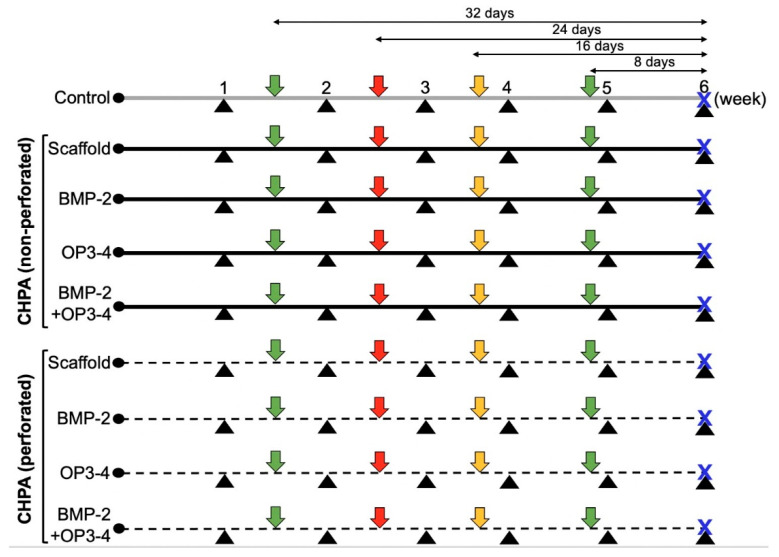
In vivo protocol. Black triangle: in vivo µCT every week; green arrow: fluorescence calcein injection 32 and 8 days before euthanization; red arrow: fluorescence alizarin injection 24 days before euthanization; yellow arrow: fluorescence demeclocycline injection 16 days before euthanization; blue X: euthanization day. Ctr: control group; CHPA: Scaffold material + Vehicle; CHPA + BMP-2: Scaffold material + BMP-2; CHPA + OP3-4: Scaffold material + OP3-4; CHPA + BMP-2 + OP3-4: Scaffold material + BMP-2 + OP3-4.

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
