# Peer review of "Perforated Hydrogels Consisting of Cholesterol-Bearing Pullulan (CHP) Nanogels: A Newly Designed Scaffold for Bone Regeneration Induced by RANKL-Binding Peptides and BMP-2"

_ijms, 2022, doi:10.3390/ijms23147768_

Round 1
Reviewer 1 Report
Well performed study and few comments.
How can you can only n=4 and still show so many significant data? Did none of the animal die during or after surgery?
Author Response
Thank you very much for your comment.
The minimum number of mice for our experiment was determined using the following formula based on a previous study where similar comparisons were performed [Sugamori et al., Bioessays. 2016].
The following text has been added in the statistical analysis subsection of the Materials and Methods.
(lines 446-456, page 12)
Sample size calculations were conducted using the formula described below:
N = 2 (Eα/2 + Eβ)2 SD2 /φ2: the minimum number of specimens that was expected to obtain a significant difference in osteogenic activity between the scaffold-only group and the scaffold plus OP3-4 + BMP-2 combination group; Eα/2 = 1.96: fixed number in the case of significance = 5%; Eβ = 1.282: fixed number in the case of power = 90%; SD: standard deviation; φ: difference between the mean value; Aν1 and Aν2:
N = 2*(1.96 + 1. 282)2*(SD) 2 / (Aν1 – Aν2) 2
3.74 = 2*(1.96 + 1.282) 2*(0.43) 2 / (1.02) 2
The standard deviations and mean differences (SD = 0.43, Aν1 - Aν2 = 1.02) of the mineral apposition rate (MAR) and osteogenic activity were obtained from the results of a previous study where similar comparisons were performed [Sugamori et al., Bioessays. 2016].
Reviewer 2 Report
Ijms-1775303
Perforated cholesterol-bearing pullulan (CHP) nanogels: a novel designed scaffold for bone regeneration induced by RANKL-binding peptides and BMP-2
The manuscript describes the investigation of the osteoconductive capacity of a newly synthesized nanogel CHPA, impregnated with OP3-4 and BMP-2.
The methods are appropriate, however, the scope is narrow and the impact is low. The novelty is low since the effect of the combination of OP3-4 and BMP-2 has been studied before, and so has the use of the CHPA. The nature of the study is combinatorial: combining previous research, and incremental: studying one variation of 300-400 μm perforation.
It remains technical report quality of mere observe and report, and lacks in-depth explanation of the underlying mechanism of the peptide-enhanced (BMP)-2-induced bone formation, as well as the mechanism of aggregation suppression. English usage needs to be improved.
Hence, a major revision is needed.
Other than addressing the main weakness listed above, the authors should also address the following additional points:
1. L42-44, within 2 consecutive sentences “shown” is used twice, and the 2 sentences convey the same message. Rewrite.
2. L48 reference is missing and should be added “In general, peptides tend to aggregate and suppress their bioactivity [10.1016/j.theochem.2003.12.040; 10.1021/jp0208891]…”
3. L52, what is “chaper-one function?”
4. L48-65 contains many confusing sentences, rewrite.
5. L71 “Using….function” does not have a verb, rewrite.
6. L286, brief synthesis route needs to be described.
7. L298, how was the pore size 300-400 μm in diameter determined?
8. L317, why was the degradation only studied for 10 day and notlonger?
9. title, “novel designed” does not sound right, change to novel, or newly designed.
10. The Conclusion is rather short thus must be re-written.
Author Response
Comments from Reviewer # 2 and our responses:
Thank you very much for your efforts in reviewing our manuscript and providing your valuable comments.
We totally agree with your remarks regarding the weakness of our research.
However, we think our manuscript is very much improved following your suggestions. We have provided point-by-point responses to your comments and questions. The revised sections of the manuscript are highlighted in yellow.
Major points:
lacks in-depth explanation of the underlying mechanism of the peptide-enhanced (BMP)-2-induced bone formation, as well as the mechanism of aggregation suppression.
Response:
The main purpose of the present study was to determine the osteoconductive effects of the newly designed scaffold, perforated CHPA hydrogels when we used the receptor activator of NF-κB ligand (RANKL)-binding peptide and BMP-2 as signaling molecules for bone regeneration. However, as we wrote at the beginning, we totally agree with your comments on our work. Based on your suggestions, we have added and clarified the following two points in the Discussion to compensate for the lack of in-depth explanation of the two issues that you indicated.
- Predicted mechanism through which RANKL-binding peptide promotes BMP-2-induced bone formation
The enhancement of osteoblast differentiation and bone formation could be explained as follows: runt-related transcription factor 2 (Runx2) is an essential transcription factor for osteoblastogenesis (Komori et al., Cell. 1997). Downstream signaling of BMP-2 is known to lead the enhancement of the Runx2 expression (Katagiri et al., Cold Spring Harb. Perspect. Biol. 2022) (Explanatory diagram 1). When BMP-2 binds to the type 2 receptor, the kinase activity of the type 2 receptor activates the type 1 receptor. This activation leads to the phosphorylation of Smad1/5, then the Smad complex associated with Smad 4 is transferred to the nucleus, upregulating the expression of Runx2 (Explanatory diagram 1). On the other hand, the downstream signaling of the RANKL-binding peptide is also known to lead the enhancement of the Runx2 expression (Ikebuchi et al., Nature. 2018) (Explanatory diagram 1). We have shown that the proline-rich domain (PRD) in the intracellular region of RANKL plays a pivotal role in the downstream signaling stimulated by RANKL-binding molecules, such as RANK-containing vesicles secreted from mature osteoclasts and the RANKL-binding peptide (Ikebuchi et al., Nature. 2018). The PRD of the intracellular region of RANKL is associated with Src-family kinases (SFKs), then SFKs lead the signaling of the phosphatidylinositol 3-kinase (PI3K)-Akt-mammalian target of rapamycin complex 1 (mTORC1) axis, which has already been reported as one type of Runx2-stimulatory signaling through ribosomal protein S6 kinase beta-1 (S6K1) phosphorylation (Explanatory diagram 1). Since both BMP-2 and the RANKL-binding peptide signaling activate the expression of Runx2, both ligands could be thought to accelerate osteoblast differentiation synergistically.
Explanatory diagram 1. A synergistic mechanism of the RANKL-binding peptides and BMP-2 on osteoblast differentiation.
BMP, bone morphogenic protein; BMPR, bone morphogenic protein receptor; PRD, proline-rich domain; Smad, Small mothers against decapentaplegic; SFKs, Src-family kinases, PI3K, phosphatidylinositol 3-kinase; mTORC1, mammalian target of rapamycin complex 1; S6K1, ribosomal protein S6 kinase beta-1; Runx2, runt-related transcription factor 2
We added the aforementioned explanation in the Discussion (line 288-305, page 8-9) without including Explanatory diagram 1, since the diagram was only prepared for Reviewer 2.
(2) Evidence of CHPA nanogels suppressing aggregation of the RANKL-binding peptide, activating the peptide
A RANKL-binding peptide, W9, normally exhibits a large particle size, with a hydrodynamic radius of approximately 300 nm in phosphate buffer solution (pH 7.4) (open square in the Explanatory diagram 2) (Alles et al., Eur J Pharm Sci. 2009). However, when the peptide is combined with a peptide-scaffold material, CHP nanogel, the particle size was reduced to less than one-tenth and the particle size becomes almost the same as that of CHP nanogel, which has a hydrodynamic radius of approximately 30 nm (open circle), indicating that the peptide is well dissolved due to the chaperone function of the CHP nanogel (the closed square).
(Chaperone function: an intrinsic ability for inducing topological changes from the aggregated form to the functional form in the cytoplasm. CHP nanogel is shown to be able to change the synthesized protein to the functionally active form by conformational modifications as the chaperone works in the cytoplasm [Akiyoshi et al. 1999].)
Explanatory diagram 2. Chart of the distribution of the hydrodynamic radius of the CHP-nanogel, the W9-peptide, and the CHP-W9-complex dispersed in phosphate buffer solution (pH 7.4) at 37°C (modified from Alles et al., Eur J Pharm Sci. 2009).
We combined the responses to Specific comments 3 and 4 regarding this mechanism of preventing peptide aggregation and the explanation of the chaperone function (See the responses to specific comments 3 and 4).
The modified text was inserted at line 50-66, page 2 in the Introduction.
Specific comment 1:
L42-44, within 2 consecutive sentences “shown” is used twice, and the 2 sentences convey the same message. Rewrite.
Response:
Thank you for your comment. We have modified the text accordingly.
Original→ Receptor activator of NF-κB ligand (RANKL)-binding peptides has already been shown as a novel signaling molecule to promote bone morphogenetic protein (BMP)-2-induced bone formation [2–6]. A RANKL-binding peptide is shown to be able to enhance BMP-2 signaling and osteogenic gene expression, leading to promoting osteoblast differentiation and ectopic bone formation [7], and the RANKL-binding peptide itself has shown to stimulate bone formation in vivo [8].
Modification→ A receptor activator of NF-κB ligand (RANKL)-binding peptide has itself already been shown to be a novel ligand that is able to promote osteogenic signaling [Ikebuchi et al., Nature 2018; Kato et al., Arthritis Res Ther 2015] and enhance bone morphogenetic protein (BMP)-2-induced bone formation both in vitro and in vivo [Rashed et al., Front Cell Dev Biol. 2021; Arai et al., Eur. J. Pharmacol. 2016; Sugamori et al., Bioessays. 2016; Uehara et al., J Dent Res. 2016; Khan et al., J Oral Biosci. 2013; Furuya et al., JBC. 2013].
Specific comment 2:
reference is missing and should be added “In general, peptides tend to aggregate and suppress their bioactivity [10.1016/j.theochem.2003.12.040; 10.1021/jp0208891]”
Response:
Thank you so much for your suggestion, we added the following references in the designated place. (line 50, page 2)
Modification→ In general, peptides tend to aggregate and suppress their own bioactivity [Laczkó et al., J Mol Struct. 2003; Malavolta et al., Protein Sci. 2006; Sasaki et al., Chem Rec. 2010].
Specific comment 3:
L52, what is “chaperone function ?”
Specific comment 4:
L48-65 contains many confusing sentences, rewrite.
Response:
Thank you for your comments.
We wrote a response to the Specific comments 3 and 4 together, since the “chaperone function” was included in the paragraph that you asked us to rewrite.
Original→In general, peptides tend to aggregate and suppress their bioactivity, and the novel RANKL-binding peptide was no exception. It has been found to be aggregated in its normal state and cannot exert sufficient pharmacological action [9]. We found that this aggregation-prone nature of peptides was suppressed by encapsulating the peptides in cholesterol-bearing pullulan (CHP) nanogels [9]. These self-assembled nanogels have a chaperone function that inhibits the aggregation of the encapsulated protein or peptide, it arranges and maintains the three-dimensional structure of the protein in a functional form [10]. We have shown that CHP nanogel released the encapsulated RANKL-binding peptide slowly compared with the collagen scaffold and maintained controlled release for 28 days [11]. Also, we showed the RANKL-binding peptide released from CHP nanogels by exchanging with the surrounding plasma proteins [12]. Regarding the scaffold for the RANKL-binding peptide, we have already used CHP nanoparticles as an injectable carrier, where acryloyl group-modified cholesterol-bearing pullulan nanogels, cross-linked by pentaerythritol tetra (mercaptoethyl) polyoxyethylene. Only one injection of the RANKL-binding peptide with the injectable nanogel carrier per day was shown to be enough to inhibit a low-calcium-induced bone loss [12]. Since eight-time injections of the same peptide per day were necessary to prevent the low-calcium-induced bone loss [13], CHP nanogels were proved to have a controlled release profile of the RANKL-binding peptide.
Modification→ In general, peptides tend to aggregate and suppress their own bioactivity [Laczkó et al., J Mol Struct. 2003; Malavolta et al., Protein Sci. 2006; Sasaki et al., Chem Rec. 2010], and the novel RANKL-binding peptide was no exception. In order to solve this problem, we previously used the cholesterol-bearing pullulan (CHP) nanogel as a scaffold of the RANKL-binding peptide [Alles et al., Eur J Pharm Sci. 2009], since CHP is known to inhibit the aggregation of the encapsulated proteins utilizing the chaperone-like function [Sasaki & Akiyoshi., Chem Rec. 2010]. The chaperone function is an intrinsic function that can change the synthesized protein to the functionally active form by conformational modifications in the cytoplasm [Hartl et al., Nature Review. 2011]. CHP was found to work as an artificial molecular chaperone to assist in proper folding to make active-form proteins and peptides [Akiyoshi et al., Bioconjug Chem. 1999]. When a RANKL-binding peptide W9 was dissolved in neutralized buffer, it showed a large particle size, with a hydrodynamic radius of approximately 300-350 nm [Alles et al., Eur J Pharm Sci. 2009]. In contrast, when it was combined with CHP nanogel, the particle size was reduced to less than one-tenth of the original size, becoming almost the same size as the CHP nanogel, with a hydrodynamic radius of approximately 30 nm [Alles et al., Eur J Pharm Sci. 2009]. The function of the peptide, which inhibits bone resorption, also appeared when combined with CHP nanogel, while the peptide could not inhibit bone resorption without combination with the CHP nanogel [Sato et al., Int J Nanomedicine. 2015]. Thus, CHP nanogel has been proven to prevent the aggregation of the RANKL-binding peptide, and change the peptide to the active form.
The above text was added in the Introduction (line 50-66, page 2).
Specific comment 5:
L71 “Using….function” does not have a verb, rewrite.
Response:
Thank you for your comment, it has been modified as suggested. (line 72-75, page 2)
Original→ Using a simple freeze-thaw method, the CHPA nanogel structure with macro-sized pores such as 20-100 μm diameter-size, maintaining its molecular chaperone-like function. Those macro-sized pores enabled cells to be impregnated into the gel [16].
Modification→ A simple freeze-thaw method changed its structure to contain macro-sized pores of 20-100 μm in diameter without modifying the molecular chaperone-like function of the CHPA nanogel. These macro-sized pores enabled cells to be impregnated into the gel [Sato et al., Sci Rep. 2018].
Specific comment 6:
L286, brief synthesis route needs to be described.
Response:
Thank you very much for your comment. We have modified the paragraph to explain the details of the synthesis of nanogel by adding Supplementary figure 2, which briefly describes the synthesis route in the Materials and Methods. (line 314-320, page 9)
Original→
CHPA was synthesized as previously described [15]. Briefly, the degree of substitution of the acryloyl groups in the CHPA was determined using 1H nuclear magnetic resonance (1H-NMR) spectroscopy (400 MHz, Avance 400; Bruker bio-spin K.K., Billerica, MA USA). DMSO-d6/D2O= 9:1 (v/v), δ=0.63–2.40 (cholesterol); 2.90–4.00 (glucose H2, H3, H4, H5 and H6); 4.67 (glucose H1 (1→6)); 4.91–5.16 (glucose H1(1→4)); 5.88–6.41 (double bond of acrylate group)) The degree of substitution of cholesteryl and acryloyl groups was 1.1 cholesteryl group and 9.3 acryloyl groups per 100 glucoside units, respectively.
Modified paragraph→ CHPA was synthesized as previously described [Kinoshita et al., J Biomater Sci Polym Ed. 2020] . Briefly, CHP was dissolved in super dehydrated DMSO, followed by the addition of 4-(4,6-dimethoxy-1,3,5-triazin-2-yl)-4-methyl-morpholinium chloride (DMT-MM). Then, N,N-diisopropylethylamine, and acrylic acid (DIPEA) were added, and the mixture was stirred for 22 h. After stirring, the liquid was collected and dialyzed with MilliQ water, and DMSO for 4 days. The degree of substitution of the acryloyl groups in the CHPA was 9.3 acryloyl groups per 100 glucoside units (Supplementary Figure 2).
Supplementary Figure 2. Synthesis route of CHPA nanogels.
CHP: cholesterol-bearing pullulan nanogel
DMT-MM: 4-(4,6-dimethoxy-1,3,5-triazin-2-yl)-4-methyl-morpholinium chloride
DIPEA: N,N-diisopropylethylamine, and acrylic acid
Specific comment 7:
L298, how was the pore size 300-400 μm in diameter determined ?
Response:
Thank you for your comment. We determined the size of the perforation from evidence shown in three references (Kuboki et al., J Bone Joint Surg. 2001, Tsuruga et al., J. Biochem. 1997, Chang et al., Biomaterials. 2000).
Explanation→ The reason for our decision regarding the perforation size was as follows. A scaffold with perforations of 300-400 µm in diameter showed higher alkaline phosphatase (ALP) activity and the increased protein expression of osteocalcin, as detected by immunohistochemistry, in comparison to the 106-212 µm-perforation group (Tsuruga et al., J. Biochem. 1997). Histologically, a scaffold with perforations of 300-500 μm in diameter showed more newly formed bone along the inner surface of the perforations and enhanced angiogenesis in comparison to the 500-600 μm perforation group (Kuboki et al., J Bone Joint Surg. 2001). Furthermore, scaffold materials with 300 µm perforations had higher bone formation and the highest maximum compressive strength in comparison to the 50, 100, 500 µm-perforation groups (Chang et al., Biomaterials. 2000).
We added above explanation to the Materials and Methods (line 328-337, page 9).
Specific comment 8:
L317, why was the degradation only studied for 10 days and no longer?
Response:
Thank you for providing a valuable comment.
In our preliminary experiment, we compared the degradation rate of two CHP nanogels; CHPA and CHPOA since we wanted to know the degradation speed of CHPA, a newly synthesized nanogel in comparison to CHPOA, which has already been used as a peptide-releasing carrier for the prevention of bone resorption in our previous study (Sato et al., Int J Contr Releas., 2015).
CHPA nanogel was expected to degrade faster than CHPOA, as CHPOA has an ester bond while CHPA does not. In order to confirm the degradation profiles of these two CHP nanogels, we performed a degradation study up to 70 days. Surprisingly, CHPA nanogel degraded very quickly in the neutralized buffer in comparison to the CHPOA nanogel, as shown in Explanatory diagram 3. On day 9 after monitoring the degradation rates, the CHPA nanogels degraded by almost 70%, while the degradation rate of the CHPOA was almost 0%, even in the alkaline buffer (Explanatory diagram 4). Since we confirmed that CHPA nanogels degraded faster than CHPOA on day 10, we stopped the monitoring the CHPA, which we had continuously monitored by day 70 after starting monitoring of the degradation rate. This is why we showed the data of the degradation rate of CHPA until day 10.
Explanatory diagram 3. Time-course study of the degradation rate of CHPA and CHPOA nanogels in pH 7.4. The detailed methods for measuring the degradation rate are described in the Materials and Methods.
Explanatory diagram 4. Time-course study of the degradation rate of CHPOA nanogel in different buffer solutions at 3 pH levels (6.0, 7.4, and 8.0). The detailed methods for measuring the degradation rate are described in the Materials and Methods.
specific comment 9:
title, “novel designed” does not sound right, change to the novel, or newly designed.
Response:
Thank you for the insightful comment. We modified it as suggested.
Besides, we also changed “Perforated cholesterol-bearing pullulan (CHP) nanogels” to “Perforated hydrogels consisting of cholesterol-bearing pullulan (CHP) nanogels” since the size of the CHP nanogel is much smaller than the 300-400 µm perforation.
Original title→ Perforated cholesterol-bearing pullulan (CHP) nanogels: a novel designed scaffold for bone regeneration induced by RANKL-binding peptides and BMP-2
New title→ Perforated hydrogels consisting of cholesterol-bearing pullulan (CHP) nanogels: A newly designed scaffold for bone regeneration induced by RANKL-binding peptides and BMP-2.
Specific comment 10:
The Conclusion is rather short and thus must be re-written.
Response:
Thank you for your comment, we modified it accordingly (line 459-464, page 12-13).
Original→ 1) CHPA nanogel was found to be a suitable scaffold for local bone formation when two signaling molecules, BMP-2 and OP3-4, were combined.
2) The size of 300-400 µm perforation in CHPA enhanced the combined effects of the two signaling molecules
Modification→ In conclusion, we herein demonstrated that CHPA nanogel is a novel scaffold that could be applied for local bone regeneration in a murine calvarial bone defect model when RANKL-binding peptides, OP3-4 and BMP-2, were used as signaling molecules to stimulate bone formation. Furthermore, the newly designed 300-400 µm-perforations of CHPA scaffolds using 3D-printed molds could be a more appropriate structure for local bone formation induced by OP3-4 and BMP-2 in comparison to a non-perforated scaffold.
Round 2
Reviewer 2 Report
After extensive revision addressing points raised, the overall quality of the MS has been improved and is suitable for publication.